# Prediction of post transarterial chemoembolization MR images of hepatocellular carcinoma using spatio-temporal graph convolutional networks

Andrei Svecic[1], Rihab Mansour[2], An Tang[2,3], Samuel Kadoury [1,2¤] *

**1** Department of Computer Engineering, MedICAL, Polytechnique Montréal, Montréal, Québec, Canada, **2** CHUM Research Center, Montréal, Québec, Canada, **3** Department of Radiology, CHUM, Montréal, Québec, Canada

¤ Current address: Polytechnique Montréal, Montréal, Québec, Canada
* samuel.kadoury@polymtl.ca

**Data Availability Statement:** The data is available at a public repository (Kaggle): https://www.kaggle.com/samuelkadoury/4d-mri-liver-dataset.

## Abstract

Magnetic resonance imaging (MRI) plays a critical role in the planning and monitoring of hepatocellular carcinomas (HCC) treated with locoregional therapies, in order to assess disease progression or recurrence. Dynamic contrast-enhanced (DCE)-MRI sequences offer temporal data on tumor enhancement characteristics which has strong prognostic value. Yet, predicting follow-up DCE-MR images from which tumor enhancement and viability can be measured, before treatment of HCC actually begins, remains an unsolved problem given the complexity of spatial and temporal information. We propose an approach to predict future DCE-MRI examinations following transarterial chemoembolization (TACE) by learning the spatio-temporal features related to HCC response from pre-TACE images. A novel Spatial-Temporal Discriminant Graph Neural Network (STDGNN) based on graph convolutional networks is presented. First, embeddings of viable, equivocal and non-viable HCCs are separated within a joint low-dimensional latent space, which is created using a discriminant neural network representing tumor-specific features. Spatial tumoral features from independent MRI volumes are then extracted with a structural branch, while dynamic features are extracted from the multi-phase sequence with a separate temporal branch. The model extracts spatio-temporal features by a joint minimization of the network branches. At testing, a pre-TACE diagnostic DCE-MRI is embedded on the discriminant spatio-temporal latent space, which is then translated to the follow-up domain space, thus allowing to predict the post-TACE DCE-MRI describing HCC treatment response. A dataset of 366 HCC's from liver cancer patients was used to train and test the model using DCE-MRI examinations with associated pathological outcomes, with the spatio-temporal framework yielding 93.5% classification accuracy in response identification, and generating follow-up images yielding insignificant differences in perfusion parameters compared to ground-truth post-TACE examinations.

**Funding:** This study was funded by the Canadian Institutes of Health Research (https://cihr-irsc.gc.ca/) through a grant awarded to SK (MOP-142401), and by Fonds de recherche du Québec en Santé (FRQ-S) (https://frq.gouv.qc.ca/en/) through a research scholarship awarded to AT. The funders had no role in study design, data collection and analysis, decision to publish, or preparation of the manuscript.

**Competing interests:** The authors have declared that no competing interests exist.

## Introduction

Hepatocellular carcinoma (HCC) is the sixth most common cancer worldwide and fourth most common cause of cancer-related mortality [1]. Several therapies have been developed for treating patients with this frequent malignancy, including transarterial chemoembolization (TACE) [2]. Initial or acquired resistance to systemic chemotherapy is the main determinant of patient survival. In current first-line palliative chemotherapy trials, response rates remain approximately 50% by standard radiologic criteria. The inability to accurately predict response to TACE drives the current practice of treating large numbers of patients with HCC, knowing only a fraction will benefit, while a significant fraction will endure treatment-related toxicities without benefits. Hence, there is a need to develop clinical tools anticipating response of HCC to TACE [2] before beginning treatment, with improved integration of temporal data. Dynamic contrast-enhanced magnetic resonance imaging (DCE-MRI) allows to survey the dynamic behavior during tumor enhancement as shown in Fig 1, and was demonstrated in previous studies to yield high accuracy in the prediction of tumor volume following neoadjuvant chemotherapy [3], when confronted to other imaging modalities such as radiographic imaging or ultrasound, which exhibits low contrast. Recent studies of DCE-MRI performed on HCCs have shown the ability to evaluate treatment response based on perfusion analysis [4]. Diffusion-weighted imaging has played a key role in predicting response. Manelli et al. [5] and Chung et al. [6] used serial diffusion-weighted MRI for TACE in HCC to improve prognosis, while Prajapati et al. [7] presented clinically relevant measures and their value for HCC prognosis.

In the past decade, machine learning has been used to train predictive patient response models from extracted imaging features (e.g. radiomics) and anticipating the pathological response [8, 9]. Approaches include neural networks [10], regression-based classification techniques [11] or more recently, using a combination of radiomics and deep learning [12], which have shown significant promise for computational medical imaging applications. A combination of imaging biomarkers was also shown to be predictive of pathological response to systemic chemotherapy and for assessing treatment response for rectal carcinomas [13]. Quantitative imaging features extracted from pre-treatment CT showed promise to forecast response in patients with colorectal metastases [14]. In the case of HCC, radiomics have been

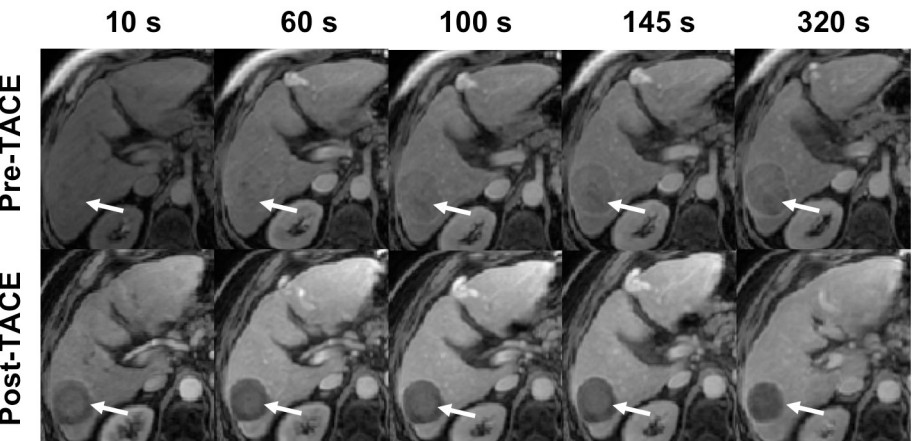

**Fig 1. Example of a dynamic contrast-enhanced (DCE) MRI sequence for assessment of HCC enhancement.** The first row presents the pre-TACE examination, while the second row presents the post-TACE examination. The white arrow indicates the location of the HCC across the arterial phases before and after treatment.

used to anticipate outcomes to specific drug treatment regimens [15, 16]. Notably, Vosshenrich et al. [17] showed a prediction of colorectal metastases with an accuracy of 0.83 and 0.96 in HCC CT images. However, predictive techniques for chemotherapy response are often based on simple linear regression, using hand-crafted features such as texture or volumetric features, without exploiting discriminative features in the temporal domain to capture changes in the hepatobiliary system. On the other hand, deep learning enables to learn low-dimensional features in an unsupervised fashion which could be used for outcome prediction [18]. Peng et al. [19] and Liu et al. [20] similarly reached promising results using deep learning-based approaches for HCC. Temporal features [21] and deep learning [22] were also used to predict outcomes, but are sensitive to limited datasets without capturing links between lesion types. More importantly, these techniques only provide a binary classification and do not produce follow-up images which can be used to extract quantifiable parameters from perfusion analyses prior to TACE.

Image prediction and image synthesis has seen a surge in popularity with the advent adversarial learning. In medical imaging, this helped to introduce the concept of domain translation to learn intrinsic relationships before different pathology contexts [23]. However, these approaches lack describing tumors in dynamic imaging data. Due to their ability to capture relations between structural components and temporal data, graph convolutional networks (GCNs) [24, 25] have been used in several applications in object identification and motion modeling, including action recognition [26], image classification [27] and tracking in video sequences [28]. GCNs were also used to describe relations between several image regions of interest (ROI) [29], but these focused solely on static representations. Furthermore, these methods do not account the inter or intra-subject morphological alterations in temporal sequences. A GCN was introduced in [30] to model the contextual interactions linking the different features from disparate ROIs, exploiting connections in salient features and global alignment. Still, these steps are computationally intensive, prohibiting an end-to-end training of the entire network, and leading to under-performing results.

Identifying responsive patients which may benefit from locoregional therapy approaches is paramount to reduce local recurrence [31]. As shown above, there is still no reliable approach that can produce accurate future image predictions of liver cancer response to treatment or assessing the possibility of local recurrence. There is a clear need to produce robust forecasting methods allowing to re-treat patients with increased drug concentrations or changing paradigms. Previous methods are based on binary classification, with no capability to predict futures images from which physiological parameters of the tumor can be measured before treatment [22]. By capturing the relationships between tumoral modifications with TACE regimens in a domain translation framework with GCNs, this would allow to build additional knowledge on patient response to chemoembolization. It allows also to configure patient-specific drug delivery which are administered during a multi-week treatment. Instead of attempting several potentially ineffective strategies, a framework attempting to learn from a cohort of previously treated liver cancer patients with HCC may elucidate which patients are more adapted to TACE [2]. This aim of this work is to provide the capability in current TACE workflows to anticipate treatment response with predicted follow-up images, where doxorubicin-based therapies may have insignificant impact on tumor viability. Furthermore, it may allow to change therapeutic strategies and thereby increasing the chances of full tumor regression.

We introduce a predictive framework to generate follow-up TACE images before treatment begins, forecasting tumoral changes based on a pre-TACE DCE-MRI and HCC annotations delineated prior to therapy. The model, named Spatial-Temporal Discriminant Graph Neural Network (STDGNN), is inspired by GCNs which have shown an ability to model relationships between structural and temporal features in contrast-enhanced imaging. A GCN is proposed

in this work to represent the intrinsic connections between image patches from time-series DCE-MRI sequences, represented as nodes in a graph. Discriminant graphs connecting these nodes are built to describe the temporal enhancement of the tumor, with the goal of offering additional data to first classify pre-TACE DCE-MRI examinations into different tumor viability classes (viable, equivocal, non-viable), which is used to drive the prediction process of post-TACE images. The motivation for introducing structural and temporal graph is to capture complementary features from both intra-phase tumor information and from individual phase acquisitions included in the 4D sequence. These features are combined with a domain translation network using concepts in optimal transport to measure the cost of mapping between different HCC viability groups in DCE-MRI. While GCN modeling has been used in several computer vision tasks, previous techniques have mostly focused solely on image-based graphs, without incorporating the structural features within image subregions to the temporal component [30].

## Materials and methods

The spatio-temporal network is first trained from a collection of pre-treatment (input) and post-treatment (output) DCE-MRI from HCCs of liver cancer patients treated with TACE. In the first step, embedded features from viable, equivocal and non-viable tumor samples in HCC are separated within a joint latent space, which is created using a discriminant graph neural network. These graphs are then used as input to the second step, where structural, temporal and global branches are implemented in the network, with the latter capturing the effort of mapping between tumor viability groups using a domain translation component based on the Wasserstein distance. In the final step, branches are fused together and fed to a decoder to produce the DCE-MRI output. For each test case, the pre-treatment DCE-MRI with HCC annotations is embedded on the discriminant latent domain, where morphological tumor response is inferred. The enhanced post-TACE with HCC viability response is generated, allowing to extract perfusion parameters in the tumoral region. The different steps of the workflow are shown in Fig 2.

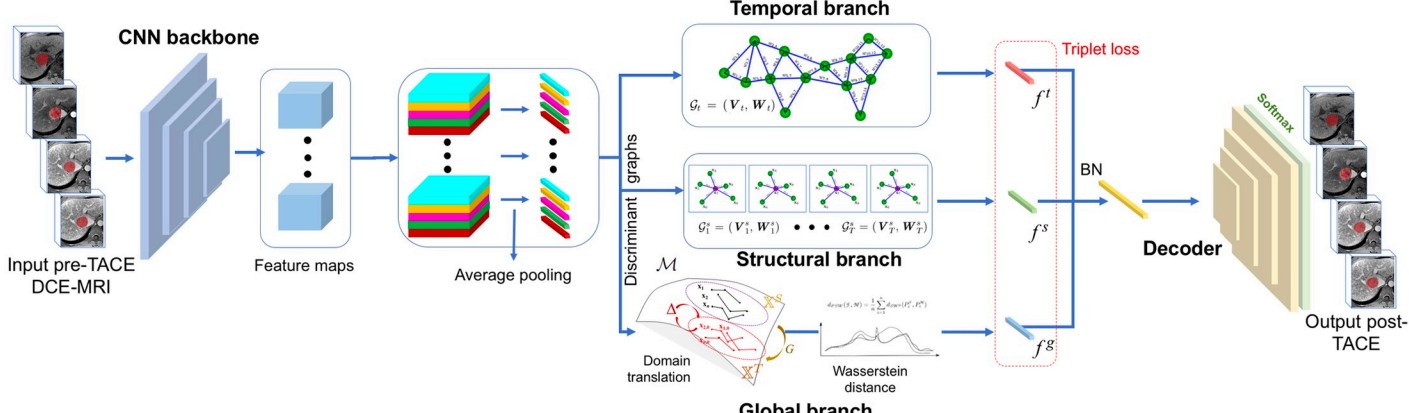

**Fig 2. Future post-TACE DCE-MRI prediction framework of HCC patients.** At training, feature maps are extracted from the baseline (pre-TACE) DCE-MRI sequences using a backbone ResNet, and projected in latent space, with data points partitioned within in a discriminant graph, with viable (V), equivocal (E) and nonviable (NV) tumor classes. Latent space samples are processed with three separate branches: a temporal branch modeling the dynamic changes, a structural branch capturing the morphological properties within single volumes, and global branch integrating an adversarial domain-translation loss, measuring the cost of mapping between source (pre-TACE $\mathbb{X}^S$) and target (post-TACE $\mathbb{X}^T$) domains in latent space $\mathcal{M}$, with the three branches combined to provide complementary features for inference. At inference time, the pre-treatment DCE-MRI is processed through a BatchNorm (BN) layer and a decoder followed by a softmax layer, to generate as an output, the follow-up sequence describing HCC response to TACE.

## Imaging acquisition

This study was performed in line with the principles of the Declaration of Helsinki. Ethics approval was granted by our Institutional Review Board (IRB) for human studies. Written informed patient consent was obtained through our IRB approved protocol to use imaging data and patient outcomes. Patients diagnosed by imaging with LI-RADS criteria demonstrating at least one tumor that is probably or definitely HCC and scheduled for TACE were chosen. A 3.0 T MRI system (Achieva TX, Philips Healthcare, Best, The Netherlands) was used for image acquisitions, with a 16-channel body array coil used for signal reception. After a gadobenate dimeglumine dose injection (MultiHance, Gd-BOPTA, Bracco Imaging SpA, Milan, Italy) administered based on body habitus (0.1 mmol/kg; maximum dose, 20 mL) and with a 1 mL/s injection, a 15 mL flush of saline was performed at 2 mL/s. After 10 s, the imaging sequence which consisted of a 4D mDixon with a 10 degree flip angle used to acquire a series of 10 breath-hold acquisitions at end-expiration, was performed 5 minutes following contrast administration. Respiratory motion correction was applied using diffeomorphic registration [32], to attenuate for tumor location variation. The field of view was $370{\times}300\text{mm}^2$, and an in-plane resolution of $1.90{\times}1.90\text{mm}^2$, with a 2.5mm spacing between slices, covering the entire liver.

## Spatial-temporal discriminant graph neural network

Given a temporally enhanced sequence, we denote $V = V_1, V_2, \cdots, V_T$ as a set of 3D DCE-MRI volumes acquired during contrast agent injection, with $T$ the number of sequences acquired during free-breathing acquisitions. For each volume of the sequence, feature maps are extracted using a backbone model, which are denoted as $F = F_1, F_2, \cdots, F_T$, with $F_i \in \mathbb{R}^{h \times w \times n}$ as the feature map of $i$-th volume, where $h, w, n$ indicates the height, width and number of slices, respectively. The entire feature map $F_i$ is partitioned into $P$ number of 3D patches, which are then treated with average pooling, yielding a normalized feature patch described as $x_i$ per patch. Therefore, for a DCE-MRI sequence with $T$ volumes, the total number of 3D patches is $N = T \times P$, with $p_i = 1, \cdots, N$ as the series of extracted patches from the dynamic sequence and associated with feature vectors $x_i$.

A spatio-temporal discriminant embedding in $d$ space is used to describe the relationship between pre- and post-TACE enhancement, using DCE-MRI sequences. We define $\mathbb{Y}^S$ as pre-TACE samples, and post-TACE samples denoted as $\mathbb{Y}^T$. Both are described in high-dimensional $D$ space. We map a dataset of patients that are viable (V), non-viable (NV) or equivocal (E) to TACE for HCC patients, which favors the distancing between sample instances using graph structures with discriminative features (see following subsection). From a radiological perspective, tumors responding to treatment are typically linked to enhancement of active edema, however it is difficult to anticipate TACE regimens based on the pre-treatment images only. Learned features from a model can help to determine the relationship with efficient drug regimens and their associated outcomes.

Provided sample 3D patches of size $5 \times 5 \times 5$ from an MRI acquired at time $t$ before treatment, $\mathbb{Y}^S = \{(\mathbf{y}_i^S, l_i, t_i)\}_{i=1}^N$ defined in $\mathbb{R}^D$ is embedded in $\mathcal{M} \in \mathbb{R}^d$, a discriminant joint manifold, with $l_i$ describing tumor viability (V / NV / E), and $t_i$ indicating the acquisition time during the DCE-MRI acquisition. With the assumption that a deep neural network can be trained on observed data lying within an underlying high-dimensional manifold, we denote $\mathbb{X}^S = \{(\mathbf{x}_i^S, l_i, t_i)\}_{i=1}^N$ existing within $\mathbb{R}^d$ and $\mathbb{X}^T$ as the target post-TACE domain. We hypothesize the existence of locally linear maps $\mathbf{M}_i \in \mathbb{R}^{D \times d}$, where local regions can be described by tangent planes $\mathbf{y}_j^S - \mathbf{y}_i^S$ and $\mathbf{x}_j^S - \mathbf{x}_i^S$, describing within linked neighbours the deviation between

pairs of points $i$, $j$. Hence, the correspondence is determined as $\mathbf{y}_j^S - \mathbf{y}_i^S \approx \mathbf{M}_i(\mathbf{x}_j^S - \mathbf{x}_i^S)$, and by assuming a global non-linearity of the underlying Riemannian manifold, a locally defined Euclidean structure can be employed.

We construct from $\mathcal{M}$ an undirected discriminant graph $\mathcal{G} = (\boldsymbol{V}, \boldsymbol{W})$, preserving the properties of local neighbourhoods in the low-dimensional embedding based on the graph characteristics captured in $\mathbb{R}^d$. The collection of nodes $\boldsymbol{V}$ (features points arising from CNN backbone and average pooling) in the graph are connected by network edges with weights $\boldsymbol{W}$ (described in following subsection). These weights are divided into feature vectors $\boldsymbol{W}_w$ and $\boldsymbol{W}_b$, representing, respectively, samples points belonging to same class clustered together and samples from different classes with similar features mapped further apart. Weights allow to compose the embedding $\mathcal{M}$. Hence, $\boldsymbol{W}_w$ and $\boldsymbol{W}_b$ partition samples in the discriminant embedding. The Laplacian matrix $\boldsymbol{L}$ is determined by $\boldsymbol{L} = \boldsymbol{Diag} - \boldsymbol{W}$, with $\boldsymbol{Diag}(i, i) = \sum_{j \neq i} \boldsymbol{W}_{ij} \forall i$ and $\boldsymbol{Diag}$ the diagonal matrix.

The relationship between input image data and generated features from the backbone model is represented with joint input-features tuples $\{\mathbf{y}_i^S, \mathbf{x}_i^S\}$. This allows a regularization on the underlying latent embedding of the generated outputs from the backbone model network, denoting the baseline DCE-MRI as $\mathbf{y}_i^S$. A softmax function $J$ assigns a probabilistic distribution of the input data $\mathbf{x}^S$. This represents the probability of belonging to either one of the response classes (N/NV/E). Values are represented as negative log-probabilities. This allows enforcing regularization of joint features to determine the associated weights, similar to a smoothing method performed on latent points based on [33]. By using minimizing iteratively the difference between edge weights $\boldsymbol{W}$ and latent samples, it avoids an overfit of the latent space dimensionality in $\mathcal{M}$:

$$\min_{\boldsymbol{W}, \ \mathcal{M}} J(\boldsymbol{W}) + \int_{\mathcal{M}} \dim(\mathcal{M}(\mathbf{y})) d\mathbf{y}$$

$$\text{s.t.} \ \ \{\mathbf{y}_i^S, \mathbf{x}_i^S\}_{i=1}^N \subset \mathcal{M},$$

(1)

with each low-dimensional point coordinate $\mathbf{y} \in \mathcal{M} = \cup_{l=1}^L \mathcal{M}_l$, $\mathcal{M}(\mathbf{y})$ describing a sub-domain associated to $\mathbf{g}$, while $|\mathcal{M}|$ an instance of $\mathcal{M}$ volumes, combining each sub-domain $L$ in the entire manifold. The regularization term ensures that the mapped input-feature tuples are mapped together within $\mathcal{M}$, of intrinsic dimensionality determined by $\dim()$ in Eq (1).

**Discriminant graph structure.** The discriminant graph structure as proposed here includes two separate sub-networks. These sub-networks provide similar features vectors on how the sample points originating from different responsive classes are distanced apart from each other within projected latent spaces. We now define the process during the training of the manifold-regularized network of determining and assigning weights to the various edge components of the graph. With every data point belonging to either one of a similarity graph $(\boldsymbol{W}_w, \boldsymbol{W}_b)$, samples linked to a particular response group (viable, equivocal or nonviable) share graph edges included in their own structure $\mathcal{G}$. Subsequently, only the latent point coordinates belonging to a particular group are used to reconstruct each sample point, where coefficients of each neighbour is calculated based on the geodesic distance within the $\boldsymbol{W}_w$ graph:

$$W_{w_{i,j}} = \begin{cases} 1 & \text{if} \ \ \mathbf{y}_i^S \in \mathcal{N}_w(\mathbf{y}_j^S) \ \text{or} \ \mathbf{y}_j^S \in \mathcal{N}_w(\mathbf{y}_i^S) \\ 0, & \text{otherwise.} \end{cases}$$

(2)

where samples within a group and between groups is given by hypersphere $\mathcal{N}_w$ defined in $d$ space, respectively, which radii are defined from the number significant weights. Similar sample points are based on weights $\boldsymbol{W}_b$ given their association to responsive or non-responsive to

TACE such as:

$$W_{b_{i,j}} = \begin{cases} 1 & \text{if } \mathbf{y}_i^s \in \mathcal{N}_b(\mathbf{y}_j^s) \text{ or } \mathbf{y}_j^s \in \mathcal{N}_b(\mathbf{y}_i^s) \\ 0, & \text{otherwise} \end{cases} \tag{3}$$

where for each $i$ sample, the closest points from other groups are included in the hypershpere $\mathcal{N}_b$. The target is to accomplish a mapping $\mathcal{M}$ in $\mathbb{R}^d$, i.e. $\mathbf{y}_i^s \rightarrow \mathbf{x}_i^s$ which is obtained during the training of the manifold-regularized graph structure. This process modifies the sample data within groups $W_w$ by drawing their location closer to the center of the assigned groups within the subspace, while distancing as much as possible the samples points with opposing treatment response as defined with neighbourhoods $W_b$.

**Temporal branch.** As initially stated, patches of different volumes from dynamic sequences can give complementary features which can alleviate several challenges such as motion artefacts and contrast agent variations. Based on the work in [30], the temporal graph structure is proposed to encapsulate relations between particular tumor patches and surrounding tissue within a the sequence of dynamically contrast enhanced volumes. Fig 3 shows the DCE-MRI sequence with $N$ patches extracted used to build the discriminative temporal graph structure such that $\mathcal{G}_t = (\mathbf{V}_t, \mathbf{W}_t)$ with $\mathbf{V}_t = \{x_1, x_2, \cdots, x_N\}$, and the graph weight matrices $\hat{W}_b^t$ and $\hat{W}_w^t$.

Using the discriminant weight matrices, the temporal branch of the model applies the temporal graph structure using the weight graphs to represent the time relationships between the different nodes extracted from the free breathing sequence. Here, $K$-layer graph convolutions are used, with the $k$-th layer included in one of the $K$ layers, and is implemented such as:

$$\mathbf{X}^k = \hat{W}_b^t \mathbf{X}^{k-1} \mathbf{B}^k - \hat{W}_w^t \mathbf{X}^{k-1} \mathbf{B}^k \tag{4}$$

with $\mathbf{X}^{(k)} \in \mathbb{R}^{K \times d_k}$ as the network features extracted in layer $k$ from the patches ($\mathbf{X}^{(0)}$ as the initial features generated by a baseline CNN), $d_k$ as the feature dimensionality and $\mathbf{B}^k \in \mathbb{R}^{d_k \times d_k}$ are the learned network parameters. A BatchNorm and a LeakyReLU is added after each layer

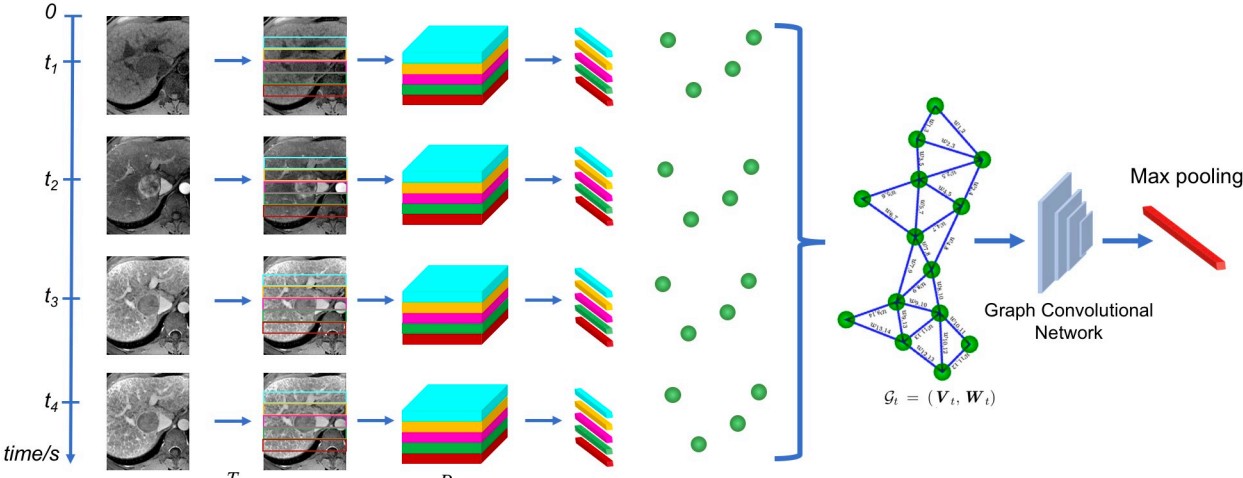

**Fig 3. Schematic representation of the the temporal branch, where a series of patches are extracted from the input volumes and features partitioned to generate a total of $T \times P$ patches for a particular input sequence.** Graph nodes are constructed from temporal feature patches, providing a representation of the entire sequence such as $\mathcal{G}_t = (\mathbf{V}_t, \mathbf{W}_t)$, which are processed with a discriminant graph network, followed by a max pool operation to obtain the feature vector.

in order to improve stabilization in training. Finally, to improve the effectiveness in the training, residual shortcuts are also added such as:

$$\mathbf{X}^k = \mathbf{X}^k + \mathbf{X}^{k-1}, 2 \le k \le K. \tag{5}$$

Following a max pooling operation, the output of the temporal branch for each sequence after the convolutional graph is denoted as $\mathbf{X}^{(K)} \in \mathbb{R}^{K^{t \times d_k}}$. We therefore generate the temporal vector $f_t \in \mathbb{R}^{1 \times d_k}$, which dimensionality is set to 512 based on the temporal resolution of the DCE-MRI examinations.

**Structural branch.** An important difficulty in tumoral characterization and contrast-enhanced dynamic imaging lies in accurately differentiating similar enhancement patterns between patients, with most previous methods using low-level features to capture tumor appearance. On the other hand, for tumor characterization from temporal DCE-MRI volumes, tissue information of the same patients will be accurate and complete since each sequence is composed of several frames covering an increased amount of data samples. Additional discriminative features can be provided from structural tumor information which can improve HCC characterization for outcome measures. Fig 4 presents the structural module, which differs from the temporal branch of the GCN.

While in the temporal branch, sequential images are used to extract tumor patches in order to build a temporal graph which encapsulates correlated features from patches throughout the time sequence, the structural branch focuses on spatial relationships between different tumor regions within individual volumes. With each volume linked to a GCN, features are then combined together to obtain an intrinsic structural feature from the DCE-MRI sequence. Hence, provided a series of $T$ volumes, the graph structure of the $i$-th volume is given by $\mathcal{G}_i^s = (\boldsymbol{V}_i^s, \boldsymbol{W}_i^s)$ with $\boldsymbol{V}_i^s = \{\mathbf{y}_{i,1}, \mathbf{y}_{i,2}, \cdots, \mathbf{y}_{i,P}\}$, and each volume divided into $P$ feature nodes, with $i$ as the $i$-th volume.

Similarly to the temporal graph branch, the associated discriminative weight matrices $\hat{\boldsymbol{W}}_b^s$ and $\hat{\boldsymbol{W}}_w^s$ are associated to each $\mathcal{G}_i^s$. Hence, an $M$-layer convolutional graph is constructed for

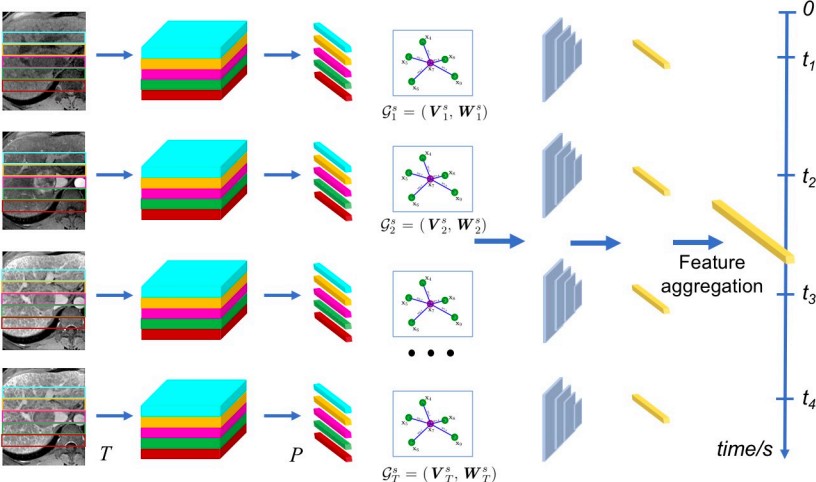

**Fig 4. Schematic representation of the structural branch, where spatial relations between extracted patches from the single enhanced volumes are exploited in order to represent the morphological features from the DCE-MRI sequence.** Graph nodes stemming from single volume feature patches are combined together in the discriminant graph network to form the feature vector of the structural data.

each volume, and for each layer $m$ ($1 \leq m \leq M$), the extracted features are obtained as:

$$\mathbf{X}_i^m = \hat{W}_b^s \mathbf{X}_i^{m-1} \mathbf{B}_i^m - \hat{W}_w^s \mathbf{X}_i^{m-1} \mathbf{B}_i^m. \tag{6}$$

Here, $\mathbf{B}_i^m \in \mathbb{R}^{d_m \times d_m}$ as the learned network parameters, and $d_m$ is the feature dimensionality, which are reduced for every individual graph structure, yielding an output of feature vector $X_i^m \in \mathbb{R}^{P \times 256}$. This is followed by a max pooling layer to reduce feature dimensionality to 256. Spatial features are then grouped together, given the overall final feature $f$.

**Domain translation branch with optimal transport.** Provided the data samples $\mathbf{y}_i^S$ in latent space, the third branch addresses the global tumor appearance as an unsupervised domain translation of HCC viability groups, based on tumor assessment on post-treatment images. It achieves domain translation of feature points in the latent domain, projecting trained feature vectors representing the overall tumor appearance to target tumor volumes at follow-up $\mathbf{x}_i^T$ using multi-domain adaptation based on the treatment response class. The model translates the baseline volumes to post-TACE tumor appearance domain based on a co-training approach which was previously proposed for organ segmentation [34]. We propose a method based on the discriminant graphs to yield with high confidence joint output features using overlapping salient characteristics from input and target domains, namely the pre-treatment DCE-MRI. A generator $G$ performing the extraction of features from DCE-MRI volumes is included in this third branch. Features are extracted based on the discriminant graph embedding as previously shown, which gives distinct higher-order forecasting features originating from the class predictions. Conversely, the deviation in output predictions is used to provide a weighting on the adversarial loss from the entire set of features, while allowing to generate prediction coherent with the overall tumor class. Lastly, $\Delta$ represents the discriminator used at the network's backend which reduces the difference in actual and predicted tumoral appearance. From the baseline domain $\mathbb{Y}^S$, temporal forecasting of the tumor response allows to create tumor volume and adversarial losses, using the features obtained from $\Delta$. Resulting forecasted tumor evolution are given to $\Delta$ once in the target space, creating directly the adversarial projection in $\mathbb{X}^T$. The network implementation is presented in the following section.

A three-term loss function is used to achieve domain translation, integrating a tumor overlapping term for the longitudinal prediction, an adversarial term which imposes a global output appearance consistent with the mappings of $\Delta$ and $G$, and a term to measure differences in graph weights. The tumor overlap term seeks to train $G$, capturing the information from the discriminant latent space knowledge which maps the baseline tumor annotation $\mathbb{Y}^S$ to the forecast volume in $\mathbb{Y}^T$, and seeks to maximize the overlap in tumor estimation based on a multi-class cross-entropy term, using a supervised approach: $\mathcal{E}_{tumor}(G) = \sum_{i=1}^D -\mathbf{M}_i \mathbf{x}_i^T \log\ p_{i,l}$, using every $D$ samples, with $p_{i,l}$ as the probability distribution that $i$ is linked to a class $l$ (V, NV, E), and $\mathbf{M}_i \mathbf{x}_i^T$ represents the baseline probability of the sample point $i$.

We incorporate in the unsupervised domain translation loss an adversarial term based on optimal transport, where $G$ is trained with dense deformations which are confronted to the deformations estimate by $\Delta$, which enables to distinguish between data samples in both domains. This objective is achieved through an adversarial loss, measuring the cost of mapping features between latent spaces:

$$\mathcal{E}_{adv}(G, \Delta) = -E[\log(\Delta(G(\mathbb{Y}^S)))] - E[\log(1 - \Delta(G(\mathbb{Y}^T)))] \tag{7}$$

and $E$ presenting the distribution's statistical expectation. We extent this loss to include an optimal transport measure based on a sliced Wasserstein distance implementation [35], by

measuring the cost of image transformation of mapping samples between the different viability group domains, and vice-versa, such that:

$$
\begin{aligned}
\mathcal{E}_{adv} \quad (G, \Delta) = & -E[\log(\Delta(G(\mathbb{Y}^S)))] \\
& -E[\phi c(p_{i,V}, p_{i,NV}, p_{i,E}) + \xi] \, \log(1 - \Delta(G(\mathbb{Y}^T)))]
\end{aligned}
\tag{8}
$$

using the optimal transport map $c$ which involves estimating the effort of mapping samples between viable $p_{i,V}$, equivocal $p_{i,E}$ and non-viable $p_{i,V}$ distributions, with $\xi$ the self-stabilizing term and $\phi$ as the weighting term. Lastly, we impose that weights stemming from distinct viability groups are indeed divergent within the latent space using the cosine distance which is minimized. The discrepancy in weights is represented as the following:

$$
\mathcal{E}_{weight}(G) = (\vec{W}_b \cdot \vec{W}_w)/(\|\vec{W}_b\| \|\vec{W}_w\|)
\tag{9}
$$

using $(\vec{W}_b, \vec{W}_w)$ which is obtained by the combination of the weights of the graph $\mathcal{G}$. We integrate the previous three loss terms within the overall function of the mapping:

$$
\mathcal{E}_{triplet}^{global}(G, \Delta) = \mathcal{E}_{tumor}(G) + \lambda_w \mathcal{E}_{weight}(G) + \lambda_a \mathcal{E}_{adv}(G, \Delta)
\tag{10}
$$

using $\lambda_w$ and $\lambda_a$ to weight the importance of the different losses (details in Network Implementation). The proposed loss function $\mathcal{E}_{triplet}^{global}$ adopts a co-training strategy, focusing on learning semantic features which are invariant to the domain, in contrast to learning specific elements associated to the domain nature, for example as morphological variations. This allows to improve the training process of the adversarial model by translating category-based features, and consequently improves the network's capability to adapt to various tumor response profiles.

## Overall loss functions

The proposed network includes a temporal, structural and global branch, with the later capturing the overall changes in tumor appearance and acting as a regularization term using a domain translation cost based on an optimal transport measure. The temporal branch captures the links between serial patches extracted from the tumor to learn dynamic information, while the structural branch encapsulates variations in tumor regions across patients.

The triplet loss function with a hard constraint on the batches is denoted as $\mathcal{E}_{triplet}$, with a softmax cross-entropy term $\mathcal{E}_{softmax}$ used to help in the training of the network. Hence, the various triplet components are combined together for the loss function:

$$
\mathcal{E}_{overall} = \mathcal{E}_{triplet}^{global} + \mathcal{E}_{triplet}^{t} + \mathcal{E}_{triplet}^{s} + \mathcal{E}_{softmax}
\tag{11}
$$

with $\mathcal{E}_{softmax} = [f^{global}, f^t, f^s]$ the softmax function combining the different features from the temporal and structural branches, and [] performing a concatenation of the features.

## Network implementation

The framework is based on a ResNet50 [36] backbone, with 3 convolutional layers (CL) of $5 \times 5$ kernels, 2 max pooling layers, a fully connected layer and a softmax layer at the end, with a stride of 1. The ResNet backbone was pretrained with ImageNet, with a random sampling strategy used to extract frames from the temporal dynamic sequences. A random sampling approach was used with $T = 4$ from every sequence, using a data augmentation strategy based on non-linear warping. For the graph convolution networks, the number of layers of the temporal module was set at 3, and at 2 for the structural branch. These were determined based on

experiments which progressively increased the graph layers, yielding a compromise between network depth and capturing sufficient features. For the domain adaptation branch with the Wasserstein distance, the backbone of the generator was a ResNet101. The discriminator $\Delta$ was configured based on a Cycle GAN architecture [37], which includes a series of 5 CL of $4 \times 4$ kernels. The channel size was set at {64, 128, 256, 512}, with ReLU functions linked with 2 strides. The final decoder includes 4 CL followed by a softmax layer. Hyperparameters were determined using an Adam optimizer, with $5e-5$ as the learning rate for the first 600 epochs, decayed by 10 at each following 50 epochs, using a momentum = 0.09 and weight decay = 0.00050. The parameters for the domain adaptation module's adversarial loss function was as follows: {$\lambda_w = 0.2$, $\lambda_a = 0.5$, $\phi = 30$, $\xi = 0.35$}. A total of 4 NVIDIA Titan X GPU K80 dual-GPU graphics cards were used to train and test the models.

## Statistical analysis

Statistical analyses were performed by a biostatistician (23 years of experience) (Software Stata/IC version 14.2). Wilcoxon tests were performed for paired sample analysis. Here, $p$ values $< 0.05$ were considered significant for this study.

## Results

### HCC transarterial chemoembolization dataset

A total of 252 HCCs from 175 patients (age 63±7) were used for training, with HCC sizes ranging between 10 and 104 mm. HCCs were categorized in viable (n = 91), equivocal (n = 85) and non-viable (n = 76) classes based on post-TACE assessment. The flowchart of patient selection is shown in Fig 5. Individuals diagnosed with HCC and awaiting TACE treatment between June 2016 and June 2020 were eligible for this study (n = 175). Image acquisitions were obtained through a prospective IRB-approved study from our tertiary referral between 2016 and 2018. Treatment response was assessed on the 6-8 week follow-up examination according to the LI-RADS treatment response algorithm, with absence of recurrence or appearance of new lesions in follow-up CT or MRI acquisitions. Patients underwent a diagnostic MRI 14 days before TACE, as well as a follow-up MRI 6-8 weeks following the procedure. Patients undergoing TACE treatment with doxorubicin for tumors on the baseline MRI were selected in this study. Inclusion criteria was a minimal HCC size of 10mm as measured on the pre- and post-treatment DCE-MRI acquisitions (registered for training using diffeomorphic registration [32]), annotated by an experienced radiologist (12 years).

The training of the networks was performed with original 512 x 512 resolution, with interslice spacing between 2 and 4 mm. The testing dataset included 65 separate patients totalling 114 HCCs (age 62 ± 7), using DCE-MRI images acquired before therapy and 6 weeks following TACE (also treated with doxorubicin). A total of 40 non-viable, 39 equivocal and 35 viable were assessed at post-TACE.

An assessment of the deformable registration method [32] used to compensate for respiratory motion during free-breathing acquisitions was first performed, as these were used for training purposes. The process involved the alignment of the pre- and post-TACE images for training, and evaluation was performed based on 10 expert annotations/volume on the early arterial enhancement phase images. Fig 6 presents sample registration results between the pre- and post-TACE images. The average target registration errors after the alignment was 1.1 ± 0.4mm.

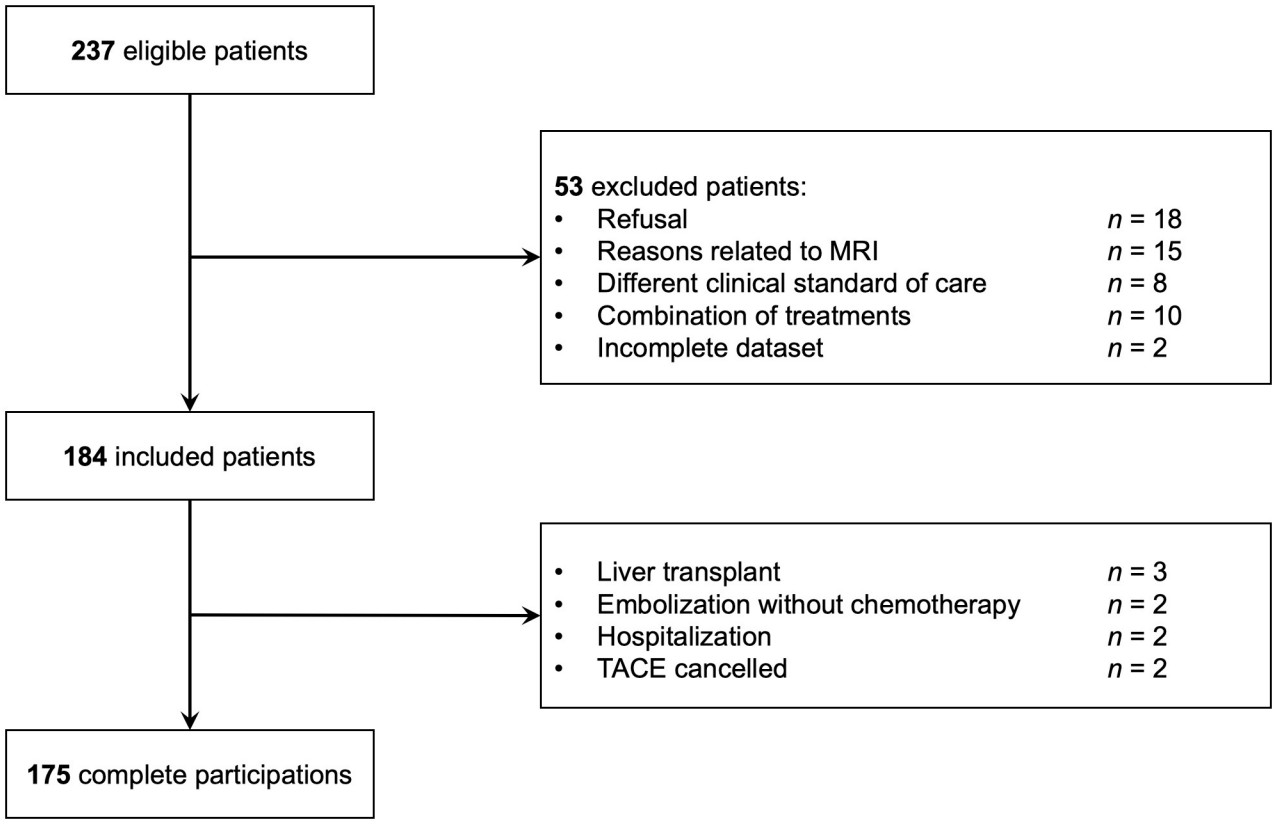

**Fig 5. Flowchart of patient selection.**

## Predicting TACE response

A five-fold cross-validation study was performed to assess the ability of the framework to preemptively identify patients responding to TACE, as this drives the future image generation process. The dataset was divided into five folds: training performed on four folds and testing on the remaining fold. This was repeated for each fold of the dataset. The separation of predicted viable, equivocal and non-viable tumors was performed based on the input baseline DCE-MRI, with temporal acquisitions at 10s, 60s, 100s, 145s, and 320s, using also the segmented HCC on the baseline exam. Findings were confirmed by fellowship-trained radiologists with expertise in liver imaging. The method was compared to two other outcome prediction methods chemotherapy procedures such as TACE. The comparative methods included one based on residual CNNs for HCC prediction (ResCNN) [19] and another based on radiomics deep learning model (RadiomicsDL) [20]. These methods were implemented to produce a 3-class classification model by adapting the output with a softmax function. The quantitative comparison with prediction results is presented in Fig 7, showing the confusion matrices. As shown in Fig 7, the STDGNN leads to an average accuracy of 93.2% from the cross-validation study, which is a statistically significant improvement to the other two methods (80.4% for ResCNN and 82.9% for RadiomicsDL) based on a paired Wilcoxon test. The comparison confronted each method to the proposed STDGNN using the HCC segmentations.

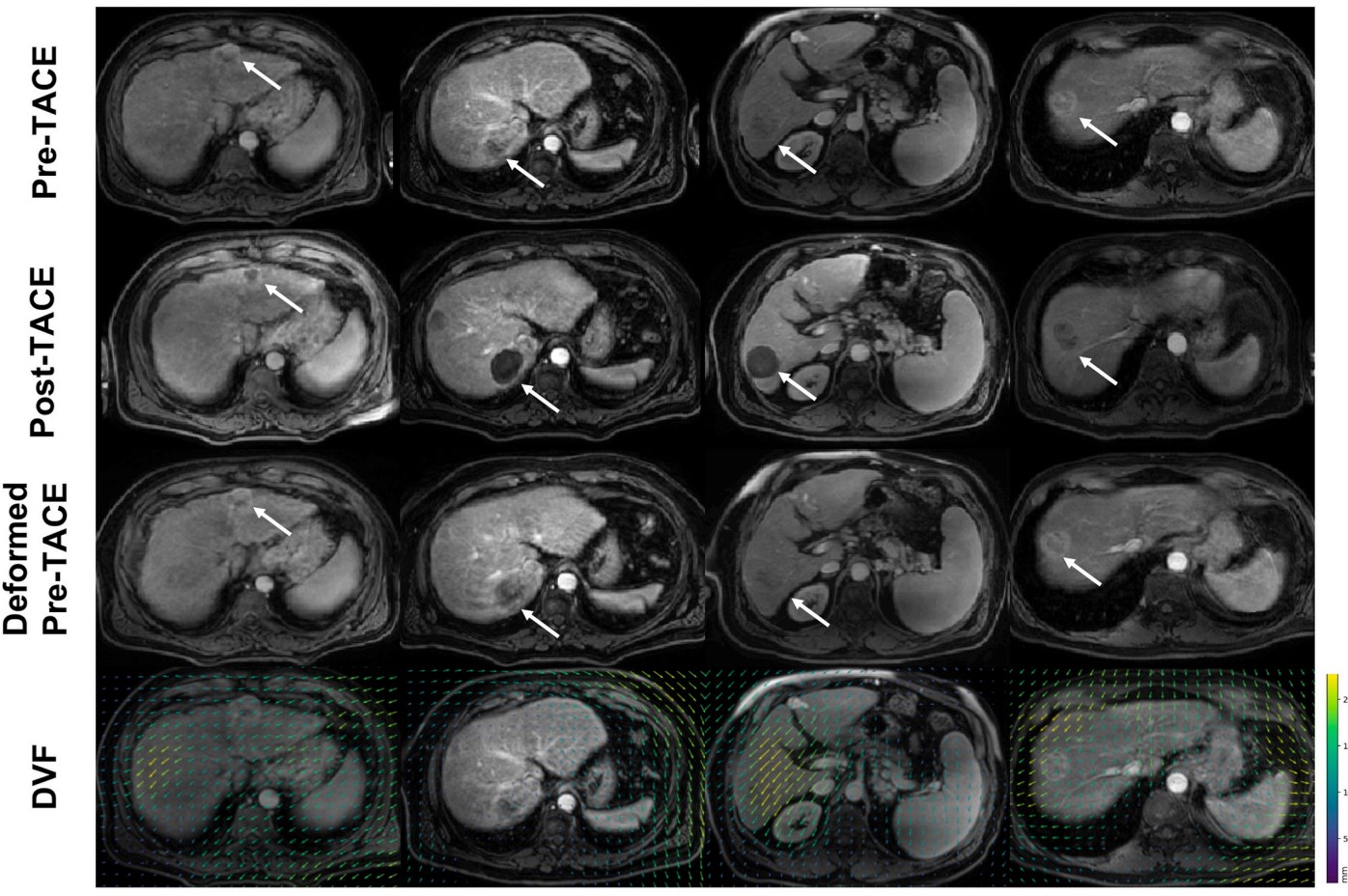

**Fig 6. Illustration of deformable registration results on 4 DCE-MRI cases (columns), achieving the non-rigid alignment of pre-TACE to post-TACE DCE-MRI examinations used for training purposes.** The last row illustrates the deformation vector fields (DVF) obtained on the coronal slices.

### Follow-up DCE-MRI evaluation

We evaluated the generation of the follow-up sequences using annotated HCCs from pre-treatment DCE-MRI scans as inputs, and where the predicted follow-up images were compared to the sequences acquired 6-8 weeks after to assess treatment response based on the radiological readings. Tumor masks were also generated on the follow-up sequence through the framework's decoder. We performed an ablation study, testing the different branch components of the STDGNN, shown in Table 1. For the predicted post-treatment HCC lesion, we evaluated the Dice coefficient and the mean squared error of the HCC surface. For the outcome, we evaluated the overall accuracy and area under the ROC curve, where the reference was the LI-RADS classification at follow-up. Results show a statistically significant improvement ($p < 0.05$) of the structural and temporal branches compared to the baseline, as well as the use of GCN compared to FCN.

Finally, we compared the predictive performance with 4 recent future image prediction methods, namely GenSeg [23], ST-ResNet [38], ST-Manifold [39] and contrastive unpaired translation (CUT [40]). The choice of these methods was made based on their ability to generate future temporal images. Table 2 compares the results in Dice and the Hausdorff distance of the HCC segmentations at post-TACE, as well as MSE in voxel intensities of the predicted

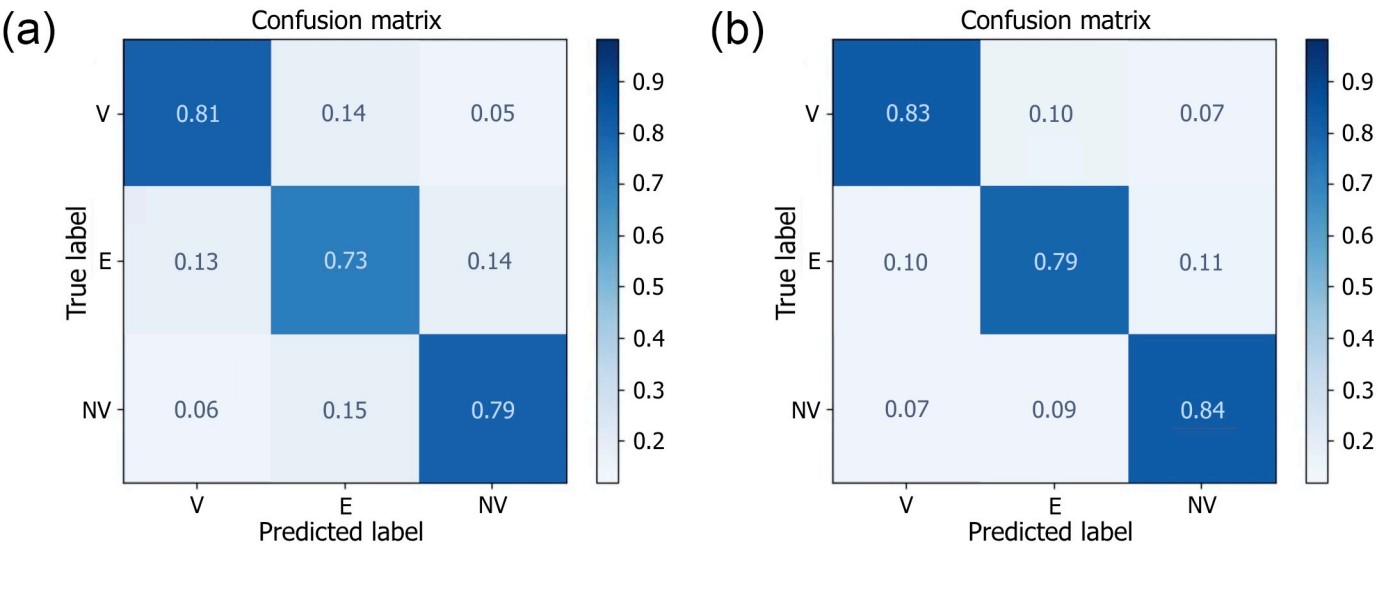

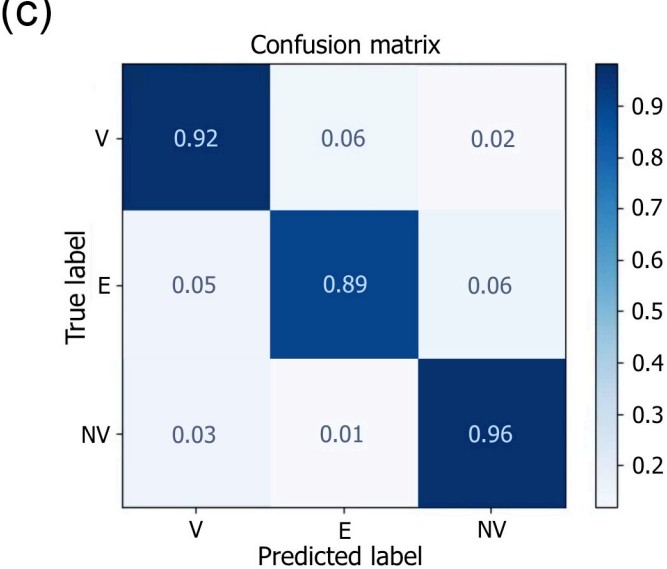

**Fig 7. Confusion matrices of tumor viability classification (V: Viable, E: Equivocal, NV: Non-viable) of HCC following TACE based on the LI-RADS score.** Inputs to each model were the annotated pre-treatment DCE-MRI acquisitions. (a) ResCNN [19]; (b) RadiomicsDL [20]; (c) Proposed STDGNN method.

images, while Fig 8 presents the box-plot diagrams of the Dice scores, stratified between viable, equivocal and non-viable lesions assessed at post-TACE. The proposed STDGNN framework yields higher Dice score in HCC tumor enhancement at follow-up (statistically significant $p < 0.05$), as compared to similar predictive frameworks. Results of the post-TACE predictions are shown in Fig 9.

## HCC perfusion analysis

The HCC characteristics from the predicted post-TACE images were evaluated with a previous dual-input single-compartment perfusion model [4]. The model allows to assess with high accuracy the portal venous input function (PIF), as well as the arterial input function (AIF),

**Table 1. Ablation experiments, comparing the addition of the structural and temporal branches to the proposed STDGNN.** A comparison is also made between the fully convolutional network (FCN) and the graph convolutional network (GCN). DSC: Dice similarity in %, MSE: Mean squared error (mm), Acc: classification accuracy in %, AUC: area under curve (AUC). Analysis of differences was performed by paired Wilcoxon tests ($p < 0.05$). Bold values indicate significant difference to baseline.

| | HCC segmentation | | Outcome | |
|---|---|---|---|---|
| | **DSC** | **MSE** | **Acc.** | **AUC** |
| Baseline (global branch) | 72.5±8.8 | 5.7±3.2 | 74.5±6.6 | 78.0±6.9 |
| Global + temporal branch (FCN) | 74.1±7.5 | 3.9±2.4 | 77.8±6.7 | 81.2±6.8 |
| Global + temporal branch (GCN) | **77.9±6.9** | **3.1±2.1** | **81.5±5.9** | **84.9±5.9** |
| Global + structural branch (FCN) | 75.6±7.3 | 3.5±2.6 | 80.2±7.3 | 82.6±6.0 |
| Global + structural branch (GCN) | **77.6±7.0** | **2.9±2.2** | **82.4±6.9** | 84.3±5.5 |
| Global + struct. (FCN) + temp. (FCN) | 79.4±6.5 | 2.7±1.9 | 83.1±6.0 | 85.2±5.7 |
| Global + struct. (FCN) + temp. (GCN) | 82.6±6.1 | 1.6±1.1 | 86.7±5.9 | 87.4±5.0 |
| Global + struct. (GCN) + temp. (FCN) | 81.8±6.6 | 1.8±1.4 | 85.7±6.5 | 86.8±4.8 |
| Global + struct. (GCN) + temp. (GCN) | **85.9±5.7** | **1.4±0.7** | **89.4±5.7** | **90.3±4.2** |

which were obtained by measuring flows in ROI identified by a radiologist in the portal veins and celiac trunk. The non-parametric analysis included measures such as the time for the contrast agent to reach its peak value (TTP), the time of arrival for the contrast agent into the tissue ($T_0$), the intensity difference (difference between baseline and maximum intensity) $\Delta S$, the normalized intensity time ratio, between TTP and peak enhancement (nMITR), the peak enhancement ratio (PER), and finally wash-out and wash-in slopes, which are calculated between the maximum and original signals. The parametric analysis included distribution volume (DV), estimating the blood flow ratio between arterial / portal plasma and the central vein, the arterial fraction (ART), as well as transfer constant from the liver tissue to the central vein ($K_2$), the transfer constant from the arterial plasma to the surrounding extravascular space ($K_a$) and the transfer constant from the portal venous plasma to the surrounding tissue ($K_p$) [41].

Fig 10 shows the enhancement variation in several regions (parenchyma, portal vein, aorta, HCC). Images were reconstructed using a motion compensation of the 4 liver motion states (from end inspiration to end expiration), which were aligned to the same reference space [32]. This allows to capture early enhancement of the HCC within the reference expiration motion state. Table 3 presents the different perfusion parameters obtained from the predicted follow-up HCC images, in comparison to the parameters extracted from the original DCE-MRI sequence. Results show that all non-parametric measures, including the nMITR, PER, TTP and wash-in / wash-out slopes yield no statistically significant difference to the ground-truth

**Table 2. Comparison with predictive methods.** DSC: Dice score coefficient of HCC segmentation, HD: Hausdorff distance (mm) of HCC segmentation, MSE: Mean squared error in voxel intensities of the predicted images. Analysis of differences was performed by paired Wilcoxon tests ($p < 0.05$). Bold values indicate significant difference.

| | Non-viable | | | Equivocal / viable | | |
|---|---|---|---|---|---|---|
| | **DSC** | **HD** | **MSE** | **DSC** | **HD** | **MSE** |
| GenSeg [23] | 66.2±8.3 | 9.2±4.3 | 0.11±0.05 | 68.5±8.4 | 9.1±4.9 | 0.09±0.03 |
| ST-ResNet [38] | 71.5±7.8 | 8.4±4.0 | 0.06±0.04 | 72.8±7.7 | 8.4±4.1 | 0.06±0.03 |
| ST-Manifold [39] | 72.4±7.3 | 8.2±3.7 | 0.05±0.02 | 73.4±7.1 | 8.3±3.7 | 0.06±0.03 |
| CUT [40] | 76.7±6.8 | 6.7±4.1 | 0.04±0.01 | 75.7±6.2 | 7.4±3.7 | 0.05±0.02 |
| STDGNN | **83.2±5.3** | **5.8±3.3** | 0.03±0.01 | **86.1±4.9** | **5.5±3.2** | 0.03±0.01 |

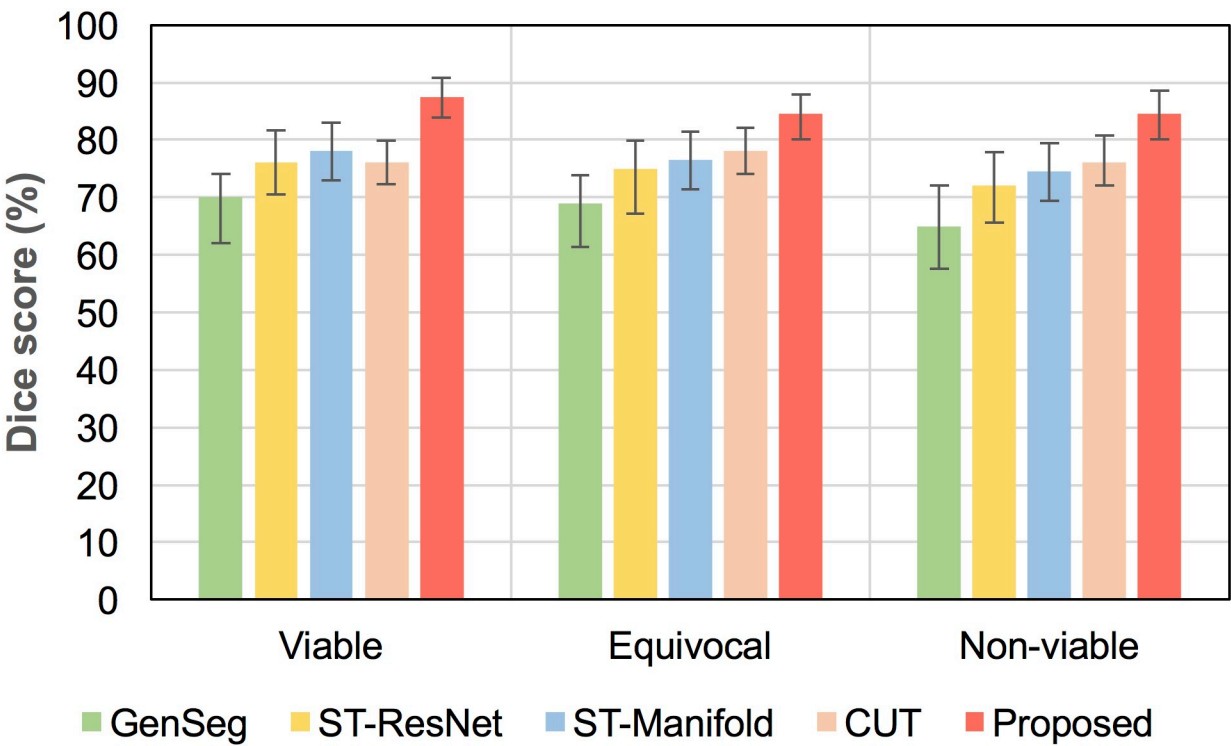

**Fig 8. Box-plots of the HCC segmentations from the predicted post-TACE images based on tumor viability class, using the original post-TACE examinations as the reference, and compared to 4 generation methods.**

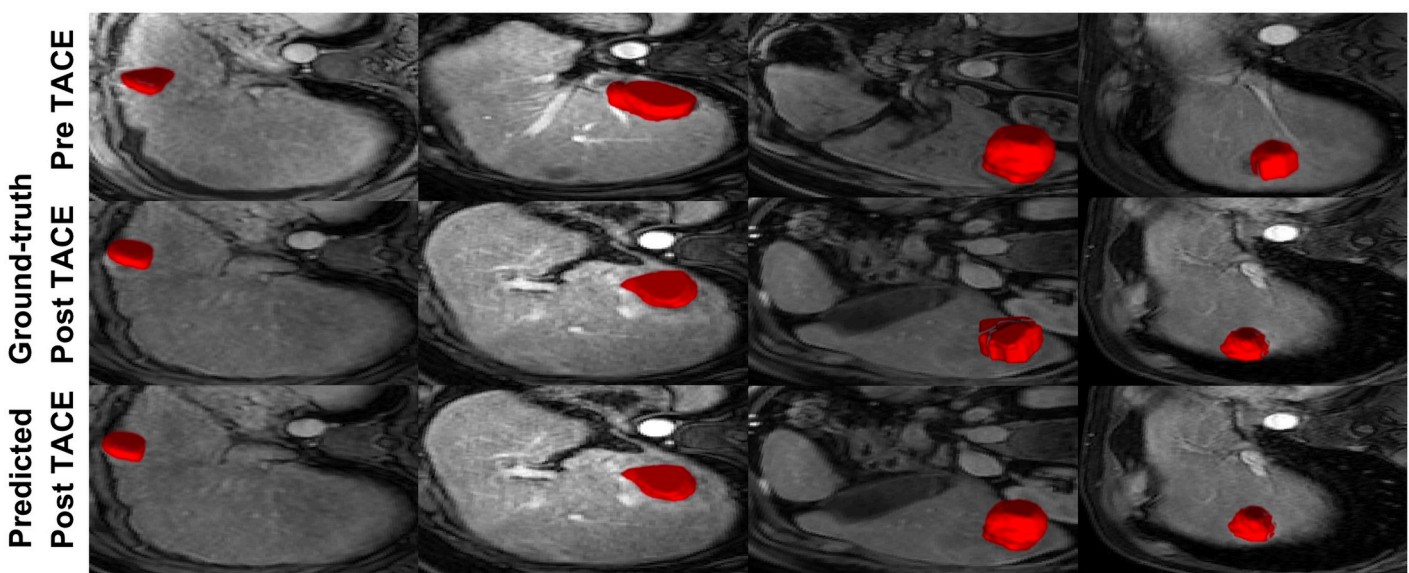

**Fig 9. Pre-TACE (used as input) and post-TACE images with HCC segmentations generated through the decoder for 4 different liver cancer patients.** The first row depicts the pre-TACE examination with HCC delineation used as the baseline input. The second row presents the ground-truth post-TACE examinations, while the third row presents the predicted post-TACE images.

(a)

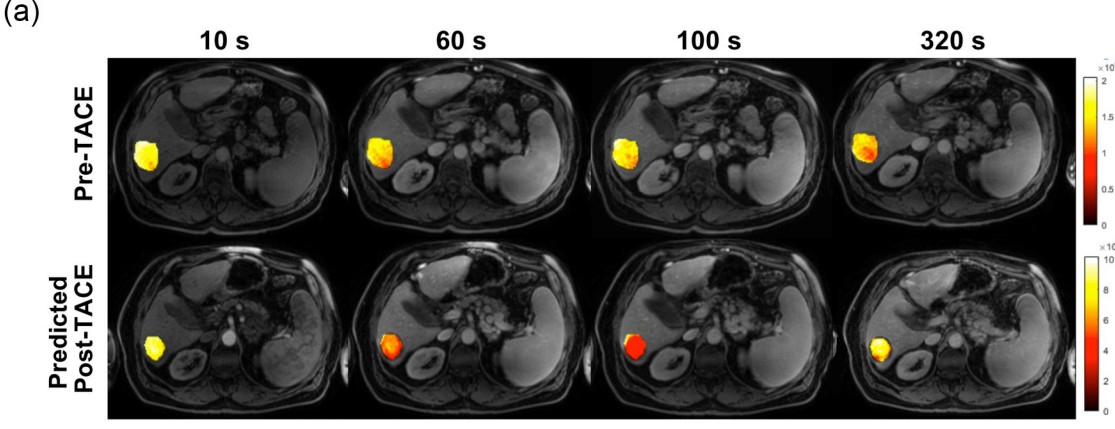

(b)

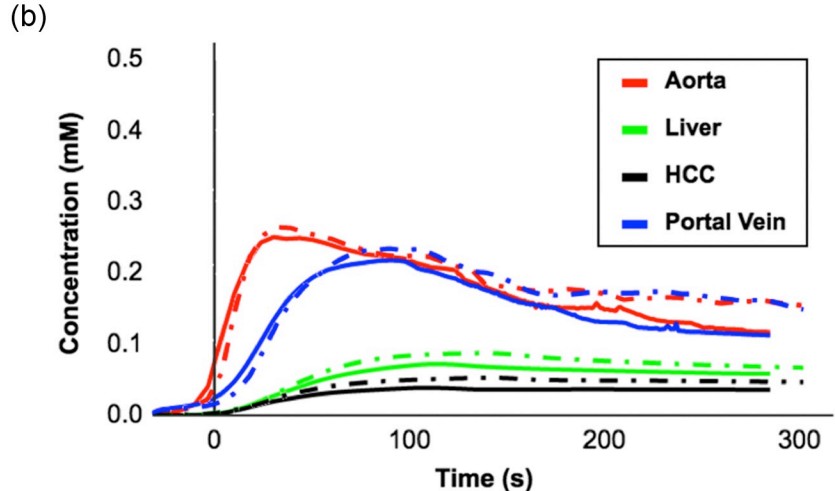

**Fig 10.** (a) Predicted tumor enhancement at sequential time points from the predicted post-TACE DCE-MRI images. Color maps indicate the changes within HCC tissue area. (b) Sample perfusion curves extracted from different liver regions (aorta, liver, HCC, portal vein), with the dashed lines indicating ground-truth post-TACE images.

sequence. As for the parametric measures, only the transfer constant $K_p$ representing the transfer from the portal venous plasma to the parenchyma, showed a significant decrease ($p < 0.05$). All other constants ($K_a$, $K_2$), as well as DV and AR, showed no difference. This can be explained variability in appearance of the liver parenchyma, which affects measures extracted outside the delineated HCC.

## Discussion

We presented a framework for the prediction of DCE-MRI images following TACE treatments of HCCs from dynamic contrast-enhanced MR imaging, to generate prior to therapy, both post-TACE DCE-MRI images and tumor region maps which can be used to assess viability. Our approach is anchored on spatio-temporal features produced from a discriminant deep neural network exploiting pre-TACE tumor viability stratification, combined with a global adversarial branch to capture tumoral changes stemming from optimal transport distance transforms. This leads to results similar to radiological interpretation based on LI-RADS

**Table 3. Parametric and non-parametric perfusion values extracted from the ground-truth (GT) and proposed predictive model (STDGNN).** Analysis of differences was performed by paired Wilcoxon tests ($p < 0.05$). Bold values indicate significant difference.

| | Non-viable | | | Equivocal or viable | | |
|---|---|---|---|---|---|---|
| | GT | STHGNN | *p* | GT | STHGNN | *p* |
| Parametric analysis | | | | | | |
| $K_p$ ($10^{-3}$) ($s^{-1}$) | 11.7±8.2 | 11.0±7.9 | **0.03** | 5.1±5.5 | 4.7±5.1 | **0.02** |
| $K_a$ ($10^{-3}$) ($s^{-1}$) | 5.6±7.6 | 5.3±7.0 | 0.60 | 6.9±10.2 | 7.0±9.8 | 0.43 |
| $K_2$ ($10^{-3}$) ($s^{-1}$) | 8.9±15.0 | 8.6±14.1 | 0.41 | 8.1±9.1 | 8.3±8.9 | 0.35 |
| DV (%) | 45.1±41.5 | 49.9±38.6 | 0.37 | 48.6±34.5 | 48.2±33.9 | 0.76 |
| ART | 0.4±0.3 | 0.5±0.3 | 0.58 | 0.5±0.3 | 0.6±0.4 | 0.47 |
| Non-parametric analysis | | | | | | |
| PER | 0.8±0.9 | 0.8±1.0 | 0.61 | 1.0± 0.5 | 1.1± 0.4 | 0.59 |
| TTP (s) | 83.3±12.8 | 83.8±13.0 | 0.88 | 69.5± 18.9 | 70.1± 19.0 | 0.32 |
| nMITR ($10^{-2}$) ($s^{-1}$) | 1.3±1.1 | 1.2±1.3 | 0.74 | 1.5±0.7 | 1.7±0.9 | 0.49 |
| Wash-in slope ($10^{-3}$) ($s^{-1}$) | 19.9±13.4 | 20.3±13.4 | 0.65 | 25.7±19.1 | 25.4±19.5 | 0.58 |
| Wash-out slope ($10^{-3}$) ($s^{-1}$) | 13.7±25.2 | 14.1±25.8 | 0.52 | 16.4±19.6 | 16.8±20.0 | 0.66 |

treatment response assessment. The input feature pairs based on a manifold regularization terms allows to establish the relationship between temporal enhancement and treatment response to a locoregional therapy such as TACE.

A significant benefit to other clinically-based criterion or machine learning methods is the ability to forecast post-TACE images with the integration of dynamic aspects of nodular arterial phase hyperenhancement via contrast injection, as well as being efficient computationally. The mean inference time for generating predictions was 3.2±0.4s, which incorporates the pre-processing time to register the images to a common reference. Furthermore, the approach yields similar perfusion analysis results to ground-truth images, from a dual-compartment parametric model representing the hepatic function [4]. The method can also be tailored to other chemotherapy procedures, where significant tumor appearance variation can be observed, as well as to other modalities such as contrast enhanced CT and to liver metastases.

This study presented three sets of experiments to analyze distinct components of the predictive framework. First, the ability of the discriminant network to segregate viable, nonviable and equivocal tumors based on the baseline examination was assessed using the tumor characteristics, as this step dictates dictate the construction of the discriminant graph network from temporally enhanced tumor volumes. Secondly, experiments were performed to evaluate future projections of tumor evolution, comparing HCC volume estimations from actual follow-up examinations, and outperforming previous spatio-temporal models used for prediction. Finally, evaluation in nonparametric and parametric perfusion analyses was performed on the follow-up examination, which provide quantitative metrics such as mean transit time and time to peak ratios. It should be noted the framework presented here allows to preemptively identify HCC's response, allowing to avoid unnecessary chemotherapy. Traditional treatment response methods rely on expert-based annotations and empirical metrics, and do not incorporate quantitative metrics capturing the dynamic nature of tumoral enhancement. The proposed method produces future post-therapy images by extracting features associated with tumor regression while integrating inter-patient variations, thereby circumventing time consuming radiological readings.

The difficulties implicated in treating patients with HCC, which is one of the most common types of cancer, are well established and properly identifying patients responding to TACE can have a major impact on planning additional embolization sessions. More importantly,

predicting tumor enhancement or increase of lesion burden is imperative to change the course of treatment. This aspect is portrayed in Fig 9, which shows an example of the forecasted HCC with an increased lesion volume compared to the segmentation in the diagnostic DCE-MRI. A prognosis framework which takes under consideration the various effects of locoregional treatments may lead to an improved clinical workflow compared to standard empirical measures. The discriminant embedding generated with the graph convolutional network (GCN) provides a mechanism to properly distinguish the variety of tumor response profiles, learning the translation between outcomes and dynamic enhancement of tumoral appearance within an adversarial domain. Even though this tool has yet to be used in the clinical setting, it provides probabilistic measures for tumor viability which may be taken under consideration when establishing a drug regimen.

Observed issues with the model was for the prediction of future post-TACE images and outcomes in cases where lesion sizes where in lower range of tumor sizes. Indeed for cases where the diameter of the HCC was below 15mm, the identification of tumor response yielding in a 11% drop in performance, which affects the overall image prediction. This is caused by the limited amount of feature patches that can be extracted from smaller volumes, as well as the reduced spatial resolution which affects smaller lesions.

The study has however some limitations. The first relates to the size of the training dataset used to create the generative model from patients treated with specific regimens of TACE. The dataset does not include the wide range of possible drug regimens and compounds, which can greatly affect the outcomes of patients, depending on tumor vascularity and viability. Even though the clinical dataset tends to be homogenous, thus helping in the model convergence from the DCE-MRI examinations pre- and post-therapy, using a more heterogenous set of patient datasets treated with different drug combinations and variation of contrast timing can help in the generalization of the model. The second issue lies in the quality of the maps used for ground-truth, which are prone to user variability. It should be noted that even though lesion maps and segmentations can be edited by radiologists, the framework is limited in incorporating variants in regimen composition or treatment course, which could be applied to each case. A final limitation lies in the model dependence on constrained timings in follow-up examinations, which can be difficult to respect for each patient, varying within 1-2 months. This makes predictions less reliable when the examinations are outside the recommended follow-up visits.

The dynamic spatio-temporal framework generated reliable follow-up exams using HCC delineated pre-treatment exams. Results are consistent with LI-RADS treatment response assessments that requires the anatomical constrains imposed by manual planning. The temporal and structural branches incorporating discriminant graph networks allow to understand the mechanisms behind patient response to locoregional treatments and why TACE regimens response vary between different physiological profiles. The framework is adaptable for lesion delineations of different sizes and margins, taking under consideration the enhancement pattern of the examined tumor, by extracting the tumoral differences in the domain translation branch. The method provides predictions that capture the temporal aspects of the examinations rather than focusing solely on specific contrast timings based on static examinations, and yields improved outcome predictions compared to recent deep learning techniques which focus on learned feature characteristics to identify tumor viability. Even though these techniques do not require manual interventions from a radiologist to select lesion features from annotations for determining treatment response, they are primarily anchored on empirical thresholds and approximate tumor delineation, thus discarding personalized modeling of lesion enhancement.

Future work will adapt the framework for liver metastases, as well as exploring native MRI data extracted directly from the k-space to train the model. We plan to establish a prospective study to assess the capability of using this predictive method during TACE planning, as well as evaluate the capability of accurately predicting patient recurrence and response. Finally, we plan to establish a multi-center study to incorporate data from several institutions to test the performance with regards to DCE-MRI acquisition variability.

## Acknowledgments

The authors would like to acknowledge William Tanguay and Catherine Huet from the Dept. of Radiology at CHUM and Guillaume Gilbert from Philips Research. We also thank NVIDIA for their donation of a GPU.

## Author Contributions

**Conceptualization:** Andrei Svecic.

**Data curation:** Rihab Mansour.

**Formal analysis:** Andrei Svecic, An Tang.

**Funding acquisition:** Samuel Kadoury.

**Methodology:** Andrei Svecic, Rihab Mansour, Samuel Kadoury.

**Supervision:** Samuel Kadoury.

**Validation:** Andrei Svecic, An Tang.

**Writing – original draft:** Andrei Svecic, An Tang, Samuel Kadoury.

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
