## [Decision Letter · Decision Letter 0]

8 Oct 2021

PONE-D-21-29673Prediction of post transarterial chemoembolization MR images of hepatocellular carcinoma using spatio-temporal graph convolutional networksPLOS ONE

Dear Dr. Kadoury,

Thank you for submitting your manuscript to PLOS ONE. After careful consideration, we feel that it has merit but does not fully meet PLOS ONE’s publication criteria as it currently stands. Therefore, we invite you to submit a revised version of the manuscript that addresses the points raised during the review process.  There was merit in your study.

Please submit your revised manuscript within 60 days. If you will need more time than this to complete your revisions, please reply to this message or contact the journal office at plosone@plos.org. Please include the following items when submitting your revised manuscript:A rebuttal letter that responds to each point raised by the academic editor and reviewer(s). You should upload this letter as a separate file labeled 'Response to Reviewers'.A marked-up copy of your manuscript that highlights changes made to the original version. You should upload this as a separate file labeled 'Revised Manuscript with Track Changes'.An unmarked version of your revised paper without tracked changes. You should upload this as a separate file labeled 'Manuscript'.

We look forward to receiving your revised manuscript.

Kind regards,

Gianfranco D. Alpini

Academic Editor

PLOS ONE

Journal Requirements:

Reviewers' comments:

Reviewer's Responses to Questions

**Comments to the Author**

1. Is the manuscript technically sound, and do the data support the conclusions?

Reviewer #1: Yes

2. Has the statistical analysis been performed appropriately and rigorously? 

Reviewer #1: I Don't Know

3. Have the authors made all data underlying the findings in their manuscript fully available?

Reviewer #1: Yes

4. Is the manuscript presented in an intelligible fashion and written in standard English?

Reviewer #1: Yes

5. Review Comments to the Author

Reviewer #1: In this study, the authors propose an approach to predict future DCE-MRI examinations following transarterial chemoem-bolization (TACE) by learning the spatio-temporal features related to HCC response from pre-TACE images. A dataset of 366 HCC's from liver cancer patients was used to train and test the model using DCE-MRI examinations with associated pathological outcomes, with the spatio-temporal framework yielding 93:5% classification accuracy in response identification, and generating follow-up images yielding insignificant differences in perfusion parameters compared to ground-truth post-TACE examinations.

1. The authors should mention the limitation of this approach in the discussion.

2. It’s unclear how the authors selectively enroll the HCC patients in this study. It’s necessary to include a figure/table to clarify this point.

3. The authors need to mention the method they used for the statistical analysis and include the significant symbol in the Tables.

4. The figure legends should be arrange in better order and not between texts.

6. PLOS authors have the option to publish the peer review history of their article (what does this mean?). If published, this will include your full peer review and any attached files.

Reviewer #1: No

---

## [Author Response · Author response to Decision Letter 0]

11 Oct 2021

First and foremost, we would like to express our sincere gratitude to the editors and reviewers, for having taken the time and effort to address comments, suggestions and questions with regards to manuscript, and improve its scientific quality. We addressed all the comments and questions raised by the reviewers and enclose the following point-by point response to those issues.

Sincerely,

The Authors

Reviewer 1:

1. The authors should mention the limitation of this approach in the discussion.

> We thank the reviewer for this suggestion. We have added the following paragraph:

“The study has however some limitations. The first relates to the size of the training dataset used to create the generative model from patients treated with specific regimens of TACE. The dataset does not include the wide range of possible drug regimens and compounds, which can greatly affect the outcomes of patients, depending on tumor vascularity and viability. Even though the clinical dataset tends to be homogenous, thus helping in the model convergence from the DCE-MRI examinations pre- and post-therapy, using a more heterogenous set of patient datasets treated with different drug combinations and variation of contrast timing can help in the generalization of the model. The second issue lies in the quality of the maps used for ground-truth, which are prone to user variability. It should be noted that even though lesion maps and segmentations can be edited by radiologists, the framework is limited in incorporating variants in regimen composition or treatment course, which could be applied to each case. A final limitation lies in the model dependence on constrained timings in follow-up examinations, which can be difficult to respect for each patient, varying within 1-2 months. This makes predictions less reliable when the examinations are outside the recommended follow-up visits.”

 2. It’s unclear how the authors selectively enroll the HCC patients in this study. It’s necessary to include a figure/table to clarify this point.

> We have added a new flowchart figure (Fig. 5) describing the patient selection workflow.

 3. The authors need to mention the method they used for the statistical analysis and include the significant symbol in the Tables.

> We have added a Statistical analysis sub-section at the end of the Methods section, detailing the Wilcoxon test (with p<0.05 indicating statistically significant differences) was used for paired statistical analysis. We added information with regards to software which was used. Finally, statistically significant differences were highlighted in bold in the tables.

 4. The figure legends should be arrange in better order and not between texts.

> This has been corrected in the revised manuscript. Figure legends are now at the end of paragraphs citing the figure, without cutting text.

---

## [Decision Letter · Decision Letter 1]

25 Oct 2021

Prediction of post transarterial chemoembolization MR images of hepatocellular carcinoma using spatio-temporal graph convolutional networks

PONE-D-21-29673R1

Dear Dr. Kadoury,

We’re pleased to inform you that your manuscript has been judged scientifically suitable for publication and will be formally accepted for publication once it meets all outstanding technical requirements.

Kind regards,

Gianfranco D. Alpini

Academic Editor

PLOS ONE

Additional Editor Comments (optional):

Reviewers' comments:

Reviewer's Responses to Questions

**Comments to the Author**

1. If the authors have adequately addressed your comments raised in a previous round of review and you feel that this manuscript is now acceptable for publication, you may indicate that here to bypass the “Comments to the Author” section, enter your conflict of interest statement in the “Confidential to Editor” section, and submit your "Accept" recommendation.

Reviewer #1: All comments have been addressed

2. Is the manuscript technically sound, and do the data support the conclusions?

Reviewer #1: (No Response)

3. Has the statistical analysis been performed appropriately and rigorously? 

Reviewer #1: (No Response)

4. Have the authors made all data underlying the findings in their manuscript fully available?

Reviewer #1: (No Response)

5. Is the manuscript presented in an intelligible fashion and written in standard English?

Reviewer #1: (No Response)

6. Review Comments to the Author

Reviewer #1: All comments for the original submission have been addressed by the authors. This manuscript is now acceptable for publication.

7. PLOS authors have the option to publish the peer review history of their article (what does this mean?). If published, this will include your full peer review and any attached files.

Reviewer #1: No

---

## [Editor Report · Acceptance letter]

17 Nov 2021

PONE-D-21-29673R1 

Prediction of post transarterial chemoembolization MR images of hepatocellular carcinoma using spatio-temporal graph convolutional networks 

Dear Dr. Kadoury:

I'm pleased to inform you that your manuscript has been deemed suitable for publication in PLOS ONE. Congratulations! Your manuscript is now with our production department. 

Kind regards, 

on behalf of

Dr. Gianfranco D. Alpini 

Academic Editor

PLOS ONE